# Assessing *Trypanosoma cruzi* Parasite Diversity through Comparative Genomics: Implications for Disease Epidemiology and Diagnostics

**DOI:** 10.3390/pathogens10020212

**Published:** 2021-02-16

**Authors:** Alicia Majeau, Laura Murphy, Claudia Herrera, Eric Dumonteil

**Affiliations:** Department of Tropical Medicine, School of Public Health and Tropical Medicine, Vector Borne Infectious Disease Research Center, Tulane University, New Orleans, LA 70112, USA; amajeau@tulane.edu (A.M.); lmurphy8@tulane.edu (L.M.)

**Keywords:** *Trypanosoma cruzi*, Chagas disease, strain diversity, polymorphism, antigen

## Abstract

Chagas disease is an important vector-borne neglected tropical disease that causes great health and economic losses. The etiological agent, *Trypanosoma cruzi*, is a protozoan parasite endemic to the Americas, comprised by important diversity, which has been suggested to contribute to poor serological diagnostic performance. Current nomenclature describes seven discrete typing units (DTUs), or lineages. We performed the first large scale analysis of *T. cruzi* diversity among 52 previously published genomes from strains covering multiple countries and parasite DTUs and assessed how different markers summarize this genetic diversity. We also examined how seven antigens currently used in commercial serologic tests are conserved across this diversity of strains. DTU structuration was confirmed at the whole-genome level, with evidence of sub-DTU diversity, associated in part to geographic structuring. We observed very comparable phylogenetic tree topographies for most of the 32 markers investigated, with clear clustering of sequences by DTU, and a few of these markers suggested some degree of intra-lineage diversity. At least three of the currently used antigens represent poorly conserved sequences, with sequences used in tests quite divergent from sequences in many strains. Most markers are well suited for estimating parasite diversity to DTU level, and a few are particularly well-suited to assess intra-DTU diversity. Analysis of antigen sequences across all strains indicates that antigenic diversity is a likely explanation for limited diagnostic performance in Central and North America.

## 1. Introduction

*Trypanosoma cruzi*, the etiological agent of Chagas disease, is a protozoan parasite that infects a wide range of mammals. It is transmitted in the feces of insects of the sub-family Triatominae during blood-feeding, congenitally, or orally via contaminated food or drink. The parasite is endemic to and widely distributed throughout the Americas; it has been reported as far south as northern Chile [1] and Argentinian Patagonia [2], and as far north in the United States as Illinois [3]. Thirty to forty percent of individuals who are infected with the parasite go on to develop chronic disease, marked by cardiomyopathies or gastrointestinal disorders such as megaesophagus or megacolon. The highest disease burden lies in Latin America and estimates suggest that around six million people may be infected globally [4]. With approximately 70 million persons at risk, Chagas disease costs 806,170 disability-adjusted life years (DALYs) annually, as well as about $627 million in patient care [5].

*Trypanosoma cruzi* presents an extensive genetic diversity [6], and based on limited sets of genetic markers, it is currently divided into seven major genetic lineages, or discrete typing units (DTUs), known as TcI through TcVI and the more recently described TcBat [7,8], which remains poorly studied using molecular markers [9]. DTUs TcV and TcVI have originated from hybridization events between TcII and TcIII, and they inherited their kinetoplast (mitochondrial) maxicircle DNA from TcIII [10,11,12]. Accordingly, TcII, TcV, and TcVI are closely related and can be difficult to differentiate with some markers, such as 24Sα [13]. DTU TcI has been the most extensively studied and it has been further subdivided into TcIa-e [14,15,16] based on sequences from the spliced-leader, or mini-exon. Indeed, a recent genome-wide analysis of TcI genomes demonstrated clear intra-DTU heterogeneity, with strong evidence of geographic clustering [17]. Genetic diversity has also been reported within TcII [18,19,20,21], and both TcIII and TcIV have been proposed to be genetically structured between North and South America [22,23].

Understanding both the full extent of the parasite’s genetic diversity and the geographical range of the various lineages is important, as strain diversity may be associated with the variety of clinical manifestations and particular parasite transmission cycles [2,7,9,14,24,25,26,27]. Phylogeography also likely plays a role in the significant discordance among serological diagnostics in some regions. Indeed, most current commercial serological diagnostic tests are based on limited sets of antigens from South American strains [28,29], which limits their diagnostic performance in North and Central America [30,31,32,33], most likely due to a failure of the antigens to react with antibodies induced by genetically diverse strains of the parasite. For example, in a study comparing three commercial and two in-house enzyme-linked immunosorbent assays (ELISAs) in Mexico, even the most sensitive of the three commercial tests only detected half of the seropositive cases confirmed by Western blot [30]. Antigen conservation for vaccine development has also been an issue, as immune selection pressure drives antigenic variation of major antigen families such as the trans-sialidase [34,35], and strain specific immunity has been detected with some vaccine candidates from this family [36]. Other antigens such as 1F8/Tc24 may be more conserved and useful for vaccine or diagnostic development [37]. Furthermore, there is some evidence that certain strains may be associated with more severe outcomes of disease development [38] and drug susceptibility [39,40,41], and fully understanding the eco-epidemiology of various genetic lineages and the composition of parasite strains infecting patients [42] may allow for more precise estimates of risk to human health and patient prognosis.

The parasite is predominantly described as diploid, with recently reported haploid genomes of 55 Mb [43] and 79 Mb [44] in length from DTUs TcI and TcV, respectively. However, the genome is markedly plastic, with genome size and karyotypic variation between strains, irregular genetic exchange, and reports of widespread aneuploidy [45,46]. The first *T. cruzi* genome was sequenced in 2005 [47], but limited comparisons of *T. cruzi* genome sequences have been performed so far, as few additional genomes were available [48,49]. These initial comparisons confirmed that extensive portions of the genome comprise repetitive sequences, making assembly complicated [48,49,50]. A “core compartment” of the genome has been identified, comprising conserved and hypothetical conserved genes, together with a “non-syntenic disruptive compartment” comprising the major multigene families of trans-sialidases, mucin-associated surface proteins (MASPs) and mucins [48]. The genetic structure of the parasite follows a “predominant clonal evolution” pattern [51], described by significant linkage disequilibrium, overrepresented multilocus genotypes, near-clades, and Russian doll patterns corresponding to the DTU subdivisions [51,52] as well as lesser near-clades within each DTU [14,18,51]. However, evidence of sexual reproduction and recombination has recently been observed within TcI, which, together with high levels of inbreeding, maintains the apparent clonal structure [53,54] but may allow the parasite to generate strain diversity.

The recent sequencing of a growing number of parasite genomes from across the Americas now allows for extensive evaluation of *T. cruzi* genomic diversity and represents an unprecedented resource for diagnostic and vaccine development. In this study, we took advantage of these sequences to perform the first large-scale analysis of *T. cruzi* genomic diversity among countries and across parasite DTUs. This comparative genomics approach also allowed investigating how different markers summarize this genetic diversity and may be used for rapid genotyping of parasites in field studies. Finally, we also examined how some of the main antigens currently used in several commercial serologic tests are conserved across this diversity of strains, as this may affect diagnostic test performance across the Americas.

## 2. Results

### 2.1. Whole Genome Comparison

We used a data set of *T. cruzi* whole genome sequences from 52 strains (Table 1), covering a wide range of geographic origin (US, Mexico, Panama, Colombia, Chile, Venezuela, Ecuador, Bolivia, Brazil), DTUs (TcI to TcVI), and hosts (mammalian hosts and triatomine vectors). We performed full genome alignments and built maximum likelihood phylogenetic trees to assess the relationships among *T. cruzi* strains. Using a representative subset of 13 genomes covering six DTUs, we observed a clear clustering of strains according to their DTU (Figure 1A) confirming that *T. cruzi* DTU genetic structure is present at the whole genome level across all six DTUs. In addition, phylogenies based on 22 TcI genomes resulted in a strong genetic structuring, with some clustering according to the geographic origin of the strains, as previously reported by Talavera et al. [17], although not exclusively, as indicated by testing isolation by distance (Figure 1B,C, Mantel test, *p* = 0.0002, R = 0.10). For example, while TcI genome sequences from the US or Panama formed single clusters, those from Colombia or Venezuela could be found in very distinct clusters. Finally, whole genome analysis further indicated that the genome of “Bug2148” clearly belonged to TcI, and was more closely related to the Dm28c and SylvioX10 strains (Figure 1), while this strain was initially sequenced as TcV [43]. On the other hand, strain H1 from Panama was associated with the TcVI DTU cluster rather than to TcI as initially classified [17].

Sequence analysis also allowed the assembly of full kinetoplast maxicircle sequences from a large number of parasite strains, for comparison with their nuclear genomes. Phylogenetic analysis of maxicircle sequences was in agreement with the previous identification of three major clades for kinetoplast maxicircle DNA [12], with clades A and C including TcI and TcII DTUs, respectively, and clade B the other DTUs (Figure 2A). However, a further genetic structuration was also detected within clade B. Indeed, TcV and TcVI hybrid strains segregated into separate clusters, distinct from both TcIII and TcIV DTUs. 

Thus, genetic structuration according to parasite DTUs could also be observed at the level of the kinetoplast maxicircle. Furthermore, detailed analysis of TcI strains within maxicircle Clade A indicated a clear sub-structuring (Figure 2B), which mirrored in part that observed at the nuclear genome level (Figure 1B), strengthening the conclusion of the existence of genetic structuration within TcI. While there are not enough genomes of other DTUs to reach this same conclusion of genetic structuration within DTU for these lineages, there does appear to be intra-lineage diversity in DTU TcII and DTU TcVI in the maxicircle sequence analysis, again similar to what is observed at the nuclear genome level. Interestingly, analysis of the maxicircle sequences also revealed two clear cases of kinetoplast introgression for TcI strains from Texas (TD23 and Corpus Christi), kinetoplast sequences of which clustered within clade B (Figure 2A) but separate from DTUs TcIII, TcIV, TcV and TcVI, while their nuclear genomes are closely related with that from TD25 strain (Figure 1B) which harbors a Clade A kinetoplast typical of TcI DTU (Figure 2A).

### 2.2. Phylogenetic Relationships Using Single and Multiple Markers

We then assessed the ability of some previously used markers to elucidate the parasite genomic diversity. Indeed, a large number of genetic markers have been proposed over the years to describe the extent of *T. cruzi* parasite diversity [2,7,55,56,57,58], including RNA genes, multiple single copy protein coding genes, as well as several genes for hypothetical conserved proteins. However, no extensive comparison of their ability to discriminate parasite strains has been performed. We identified nucleotide sequences corresponding to 30 individual markers (Appendix A) using basic local alignment search tool (BLAST) searches of the full genome sequence dataset and GenBank sequences. Individual markers varied in size from 118 to 2260 bp, and we constructed phylogenetic trees from each individual marker based on maximum likelihood (Appendix A). We observed very comparable tree topographies for most markers, with clear clustering of sequences by DTU, except for TcV and TcVI, which often clustered with TcII and TcIII, in agreement with their hybrid nature. However, most markers including the 24S RNA, MSH2 or TcSD5D, which have been extensively used for genotyping [7,22,55,56], showed limited or no sequence variation within DTUs. A few markers such as the flagellum-adhesion glycoprotein or Beta-adaptin did, however, clearly resolve DTUs while also evidencing some level of intra-lineage variation, particularly for TcI and TcII. These markers may thus provide finer resolution for the detailed assessment of parasite genetic diversity beyond the major lineages.

We also evaluated how the mini-exon spliced leader marker discriminated among strains, as it has been one of the most extensively used marker [7,14,16,58,59,60,61,62]. BLAST analysis confirmed that it is a high copy number sequence, with up to 800 copies per genome, depending on the strain (Appendix A). We evaluated how multiple copies of the mini-exon sequences may overlap or blend among strains and DTUs. Phylogenetic analysis indicated that most copies clearly clustered for each strain and each DTU, following a Russian doll pattern, although some strains presented multiple clusters of mini-exon sequences (Figure 3). A notable exception was for a small number of sequences from TcII strains, which clustered with both TcV and TcVI sequences, in agreement with the hybrid origin of these DTUs (Figure 3B).

Detailed phylogenetic analysis using a single dominant mini-exon sequence from each strain resolved well all DTUs (Figure 4), including DTUs TcII, TcV and TcVI, which were clearly identified and well separated from TcIII (Figure 4A,B). In addition, the mini-exon revealed particular intra-lineage diversity for TcI (Figure 4C) and hinted at a similar diversity within other DTUs for which multiple sequences were available. Within TcI, this genetic diversity was mostly associated with the geographic origin of the strains with a strong isolation by distance (Figure 4D, Mantel test, *p* = 0.0002, R = 0.54). The mini-exon sequences were well suited to assess *T. cruzi* parasite diversity at the DTU level (Mantel test, *p* = 0.0001, R = 0.70) as well as within DTUs (Mantel test, *p* = 0.0002, R = 0.61) and summarized well the diversity observed above at the whole genome level. Mini-exon sequence from strain “Bug2148” also clustered with sequences from TcI DTU, in agreement with the whole genome analysis above. Similarly, the mini-exon sequences of strain H1 from Panama clustered with DTU TcVI.

We also assessed the usefulness of phylogenetic trees constructed from concatenated sequences of most of the available markers (*n* = 30 markers), corresponding to approximately 26 kbp of sequence for each strain (Figure 5). These trees showed defined clusters for all DTUs (Figure 5A), as observed for whole genomes and the mini-exon marker. However, this approach allowed identifying separate haplotypes for TcV and TcVI hybrid strains, corresponding to a TcII-like haplotype and and TcIII-like haplotype, respectively. Though the TcII-like concatenation demonstrated higher congruence (Mantel test, *p* = 0.0001, R = 0.71), both this and the TcIII-like concatenation (Mantel test, *p* = 0.0003, R = 0.59) were similar overall to the whole genome analysis. Trees from concatenated markers also confirmed that “Bug2148” clustered with TcI strains and H1 from Panama clustered with TcV/TcVI strains (Figure 5A). As in the analyses above, intra-lineage diversity could be observed based on the concatenated sequences for those DTUs for which sequences from multiple strains are available, particularly TcI and to a lesser extent TcII and TcVI. Accordingly, a separate analysis of TcI sequences (Figure 5B) confirmed a strong genetic structuration of TcI parasites rather comparable to the whole genome analysis (Mantel test, *p* = 0.0001, R = 0.75), with multiple well-supported clusters, which mostly corresponded to the geographic origin of the strains (Figure 5C, Mantel test, *p* = 0.0001, R = 0.35). Thus, the use of multiple markers provided limited additional information compared to the mini-exon sequence alone.

### 2.3. Diagnostic Antigen Conservation

The high genomic diversity of *T. cruzi*, both among DTU as well as according to their geographic origin within DTUs, supports the hypothesis that limited antigen conservation may explain the variable performance of several current commercial serological diagnostic tests in some geographic areas. To further test this hypothesis, we performed BLAST searches of the available genomes for seven antigen sequences that are a subset of antigens used in some of these tests (Table 2), to assess the conservation of their predicted translation. Antigen1, Antigen36, and KMP-11 which correspond to small repeated sequences, were the most conserved, with alignments of peptide sequences representing all DTUs demonstrating an average 95.3% pairwise identity for Antigen1, 98.3% pairwise identity for Antigen36, and 95.6% pairwise identity for KMP-11. The sequences used in diagnostic tests were most closely related to those from strains from south America and from multiple DTUs but diverged more from sequences from Central and North American strains (Appendix A).

Other antigens including the shed acute phase antigen (SAPA) repeat region, Antigen30, TcH49, and B13 presented more limited sequence conservation across all strains, with respective pairwise identities of 77.0%, 88.2%, 94.9% and 69.2%. The reference antigen currently used in serology was quite different from many sequences from other strains (Figure 6A,B). Although sequence diversity of the SAPA antigen was high, it did not seem to be structured according to geographic origin of the strains or parasite DTU, and this diversity was observed for both the repeat region of the protein that is used in diagnostic tests (Figure 6B,C) or the full length protein (Appendix A). Overall, the poor conservation of these antigens across parasite DTU and geographic diversity may explain the limited performance of tests based on these antigens in Central and North America.

## 3. Discussion

The genetic diversity of *T. cruzi* has been detected a long time ago [68,69], but understanding its full extent and the geographical ranges of the various lineages remains critical, as strain diversity may be associated with the diversity of clinical manifestations and parasite transmission cycles. Here, we performed the first extensive analysis of *T. cruzi* genomes from multiple DTUs and geographic origins, to shed light on the phylogeography of the parasite. 

A first key finding was that *T. cruzi* genetic structure corresponding to DTUs is maintained at the nuclear genome level. While this subdivision has been previously established with a small number of genetic markers [7], it is important to see it reflected at the whole genome level as this confirms that it corresponds to a strong evolutionary signal of *T. cruzi* genetic structure and strain diversity, and a useful classification of strains. However, we acknowledge that given the limitations in the number of available complete genomes from non-TcI strains, further confirmation of this result with additional TcIII and TcIV strains included is needed. In spite of a different inheritance mechanism and different evolutionary pressures, this structuration according to DTU was also reflected to some extent in the kinetoplast maxicircle sequences. Nonetheless, analysis of kinetoplast maxicircle also revealed clear examples of introgression. The close relatedness of the two strains displaying this introgression, as well as the apparently ancient origin of their kinetoplast suggests that they may have derived from a common ancestor, and it raises the question of the mechanisms and frequency of such kinetoplast introgression. Indeed, while *T. cruzi* population structure has been assumed to be mostly clonal [52,70,71], recent studies pointed to sexual reproduction and recombination at rates that reject a purely clonal model, associated with a high inbreeding [53,54]. Such sexual reproduction may provide opportunities for different inheritance patterns for kinetoplast DNA.

Within DTUs, a strong genetic structuring of *T. cruzi* was also detected within TcI, in agreement with a previous study [17]. This structuring corresponded in part to the geographic origin of the strains, but not exclusively, suggesting that multiple factors contribute to the diversity of strains within TcI. Indeed, multiple sublineages were detected within TcI from the same geographic region and possibly correspond to the previously detected groups TcIa to TcIe based on mini-exon sequences [14,15,16], which appear in distinct genomic clusters, but genomes from additional strains are needed to assess this correspondence. These multiple sublineages from the same country or region may be a result of distinct epidemiological and transmission cycles resulting from different hosts and environments. Alternatively, certain strains isolated in a particular region may represent introductions to the area as a result of host movement. Similarly, TcII, and to some extent TcVI, appear to exhibit intra-DTU diversity as well, and comparison of a greater number of strains from all DTUs will be necessary to assess their intra-DTU diversity and genetic structure.

While whole genomes certainly provide the most complete evaluation of *T. cruzi* diversity, our analysis of single and concatenated markers provides strong evidence that the extent of genomic diversity can be captured through limited sequence information. This would be particularly useful for relatively simple genotyping procedures of a very large number of strains to further study parasite diversity in endemic regions. Indeed, most of the single gene markers evaluated here, such as TcSD5D or the 24S RNA gene, can capture parasite diversity among DTU well, confirming their usefulness for such classification. Nonetheless, most of these markers are too conserved to detect significant intra-DTU diversity, and thus lead to an underestimation of parasite diversity. The mini-exon appears as a particularly good marker to assess both DTU and intra-DTU diversity. Although the polymorphic nature of the mini-exon has called into question its ability to summarize strain diversity because of intra-strain variation among orthologous copies, our results demonstrate that it recapitulates well multi-markers and whole genome diversity. Indeed, its intra- and inter-genome sequence diversity appears in agreement with the Russian doll pattern proposed earlier to describe *T. cruzi* population structure [51]. The low mini-exon copy number observed in some strains likely represented limited sequencing coverage rather than a true lack of gene copies, as the spliced leader is essential for parasite viability [72,73], again pointing to limitations with currently available data. Therefore, our results indicate that to tease out local eco-epidemiology of the parasite, it is possible to focus on limited markers for a larger number of samples rather than on a high number of markers for fewer samples. At the same time, access to a greater number of genomes or reference sequences from those DTUs and geographic regions not well-represented in the literature is crucial to fully describe intra-lineage diversity for all DTUs. 

Our analyses also indicated some unexpected discrepancies for some strains. First, H1 strain from Panama, was initially classified as belonging to TcI [17], but it rather belongs to the TcVI DTUs, and it is thus the first TcVI genome from a strain from Central America. Second, the genome of strain “Bug2148”, initially reported as belonging to TcV [43], unambiguously clustered with TcI strains for all markers tested as well as at the whole genome level. As multiple studies with different markers have previously shown that the original Bug2148 strain indeed belonged to TcV [74,75], a likely explanation is that a different strain (belonging to TcI) was actually sequenced instead of Bug2148.

With such limited degree of genome conservation across all DTUs, diagnostics are likely to result in false negatives in particular regions where genetically diverse parasite strains may be circulating. Our analysis of some of the antigens currently used in diagnostic tests indicate that several antigens are poorly conserved across various parasite strains and DTUs. Thus, these tests are suited to detect human infections with a limited diversity of DTUs and strains but may not detect infection with strain variants. This may explain the limited performance of current tests in North America [30,31,32,33] and highlights the need for new tests able to detect infections with the full diversity of *T. cruzi* parasites.

## 4. Conclusions

In conclusion, whole genome comparison confirmed the strong genetic structure of *T. cruzi* strains, in agreement with its current subdivision into well-established DTUs [76]. However, some significant intra-DTU diversity is emerging, associated in part to geographic structuring. Although not all tested markers perform equally well, most are well suited for estimating parasite diversity to DTU level, and a few such as the mini-exon, are particularly well suited to assess intra DTU diversity as well. Analysis of antigen sequences across all strains confirms that antigenic diversity is a likely explanation for limited diagnostic performance in Central and North America. Identifying new antigens conserved across the extent of diversity may lead to improved serological diagnostics.

## 5. Materials and Methods

### 5.1. Trypanosoma cruzi Genome Sequences and Markers

Genome sequences from 52 previously described parasite strains were included in this study, covering TcI to TcVI (Table 1). Notably, no whole genome sequence of TcBat is currently available.

For strains CL-Brener, Dm28c, 231, TCC, SC43, S11, Ycl4, S23b, S92a, S15, S162a, S44a, S154, G, and Bug2148 we used previously published assemblies [18,44,47,48,77,78,79]. For all the remaining strains, we used raw sequence reads from the SRA database [80]. These were newly assembled in Geneious 11.1.5, using Dm28c or TCC as reference genomes, as these represent the most completely assembled genomes to date [48]. The use of new assemblies allowed having a more homogenous dataset that minimized potential bias due to differences in sequencing methodologies and assemblies. These new genome assemblies from each strain were used to generate strain-specific databases in Geneious 11, and these were searched using the BLAST for 35 markers, most of which represent single copy sequences that had been used before for identifying *T. cruzi* genotypes and evolutionary relationships among DTUs (Appendix A). Raw sequence reads were similarly used to assemble kinetoplast maxicircle sequences, using 231 (KC987253), CLBrener (DQ343645), Esmeraldo (DQ343646) and SylvioX10 (FJ203996) maxicircle sequences as references.

### 5.2. Whole Genome Alignments

Whole genome alignments were constructed using the progressiveMauve algorithm and default settings [81]. Thirteen genomes representing all DTUs except TcBat and 22 genomes representing only TcI were separately aligned to identify homologous genome regions, referred to as locally collinear blocks (LCBs) in Mauve. The extent of LBCs was compared across genomes to assess conservation among all six DTUs, as well as among TcI genomes, and concatenated LCB sequences were used to elaborate maximum likelihood trees using PHYML as implemented in Geneious 11. Isolation by distance was tested using Mantel test as implemented in Past3 software [82] and pair-wise geographic distances among strains were based on their country of origin.

### 5.3. Single Marker Analysis

Multiple sequence alignments were made for all available sequences of each individual marker using MUSCLE as implemented in Geneious. Phylogenetic trees based on maximum likelihood as implemented in PHYML were constructed using the Phylogeny.fr tool [83] or IQ-Tree [84] for each marker. For the analysis of the mini-exon sequences, additional reference sequences were downloaded from GenBank [85] database (Tu18: AY367125.1, Esmeraldo: ANOX01015457.1, CLBrener: U57984, 9122102r: AY367124, MT4167: AF050523, CanIII: AY367123.1, TCC1122cl7: KT305876, TCC2476cl6: KT305883, M5631: AY367126, M6241: AF050522, SylvioX10: CP015667, P/217cl1: EF576840.1, S040cl1: EF576842, Raccoon70: EF576837, SC43: AY367127.1, MN: AY367128.1). As the mini-exon is a multicopy gene of approximately 600 bp with variable copy number depending on strain, any BLAST return of at least 250 bp was considered for analysis. To evaluate how multiple copies of the mini-exon sequences may overlap or blend among strains and DTUs, parasite strains with sufficient genome coverage to allow for identification of a large numbers of mini exon paralogous sequences of sufficient length were analyzed separately. Phylogenetic trees were edited using FigTree 1.4.4 for final rendering [86]. The Mantel test, as implemented in PAST software [82], was used to determine congruence between trees based on whole genomes and trees based on the mini-exon, both for all DTU trees and for TcI only trees.

### 5.4. Concatenated Markers Analysis

Concatenations were made of marker sequences for each genome/strain. Because some marker sequences were absent from certain genomes due to limited sequencing coverage, a total of 30 markers were concatenated for 44 genomes. For those DTUs that are hybrid in origin, namely TcV and TcVI, two concatenations were made: one for the TcII-like sequences and one for the TcIII-like sequences. Concatenated sequences were aligned using MUSCLE, and maximum likelihood phylogenetic trees were constructed using PHYML as for single markers. Separate alignments and phylogenetic trees were also constructed for TcI and non-TcI parasite DTUs for increased resolution of within-DTU diversity. Isolation by distance was tested using Mantel test as implemented in Past3 software [82]. The Mantel test was also used to determine congruence between whole genome trees and concatenated trees for all DTUs and TcI only. For all-DTU concatenated trees, separate iterations of the test were run for the concatenation containing the TcII-like sequences and that containing the TcIII-like sequences.

### 5.5. Antigen Conservation

To evaluate the conservation of antigens that are used in current commercial serological diagnostic tests for *T. cruzi*, genome databases were searched via BLAST for sequences corresponding to seven antigens (Table 2).

Commercial tests represented include Chagas StatPak^®^ (ChemBio Diagnostics Inc., Hauppauge, NY, USA) Trypanosoma *Detect*™ Rapid Test (InBios International Inc., Seattle, WA, USA), and Chagatest ELISA (Wiener Lab, Rosario, Argentina). StatPak^®^ utilizes antigens B13, H49, and 1F8. 1F8 was excluded from our analysis as has been extensively characterized previously [37]. Trypanosoma *Detect*™ consists of a fusion peptide made up of portions of antigens 1, 30, 36, SAPA, Kmp-11, and multiepitope recombinant TcF. As TcF epitopes did not return BLAST results in numerous genomes, this antigen was excluded from further analysis. Wiener’s Chagatest consists of antigens 1, 2, 13, 30, 36 and SAPA. Antigens 2 and 13 are very short peptide sequences (12 aa and 5 aa) that are identical to epitopes within B13 and TcD, part of the TcF multiepitope recombinant protein, so were not included separately in our analysis. DNA sequences were translated into protein sequences for each *T. cruzi* strain, using all six reading frames and selecting the translation that most closely matched the reference sequence annotations. All peptide sequences were aligned for each antigen, including the reference sequence, and trees were constructed from these alignments using PhyML with the LG (Le Gascuel) substitution model. 

## Figures and Tables

**Figure 1 pathogens-10-00212-f001:**
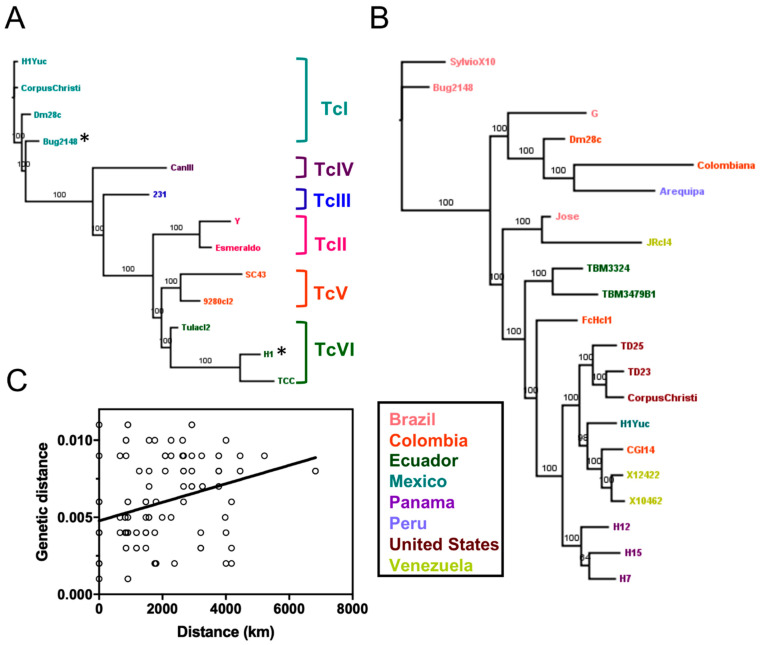
Whole genome analyses of strains representing all discrete typing units (DTUs). Phylogenetic trees constructed from whole genome alignments demonstrate a clear clustering by DTU (**A**) as well as strong sub-structuring within DTU TcI (**B**), with some geographic association (**C**) by Mantel test (*p* = 0.0002, R = 0.10). Asterisks indicate strains that were previously assigned to a different DTU.

**Figure 2 pathogens-10-00212-f002:**
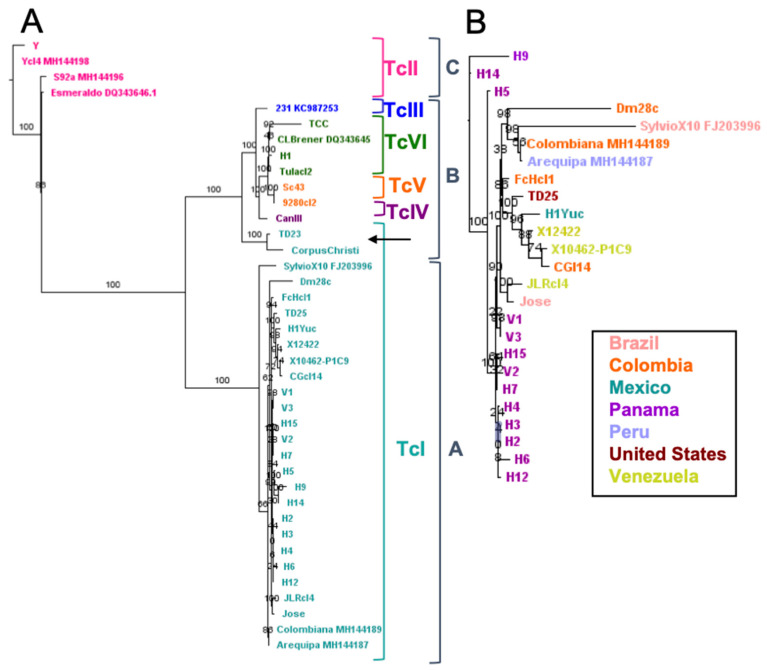
Analysis of kinetoplast maxicircle sequences. Three major clades were observed (**A**), in agreement with previous identification. Clades A and C include DTUs TcI and TcII respectively, and Clade B includes the other DTUs, with a structuration within this clade by DTU. Two clear cases of kinetoplast introgression (arrow) for TcI strains from Texas are present. Detailed analysis of TcI strains (**B**) indicated sub-structuring, similarly in part to the nuclear genome.

**Figure 3 pathogens-10-00212-f003:**
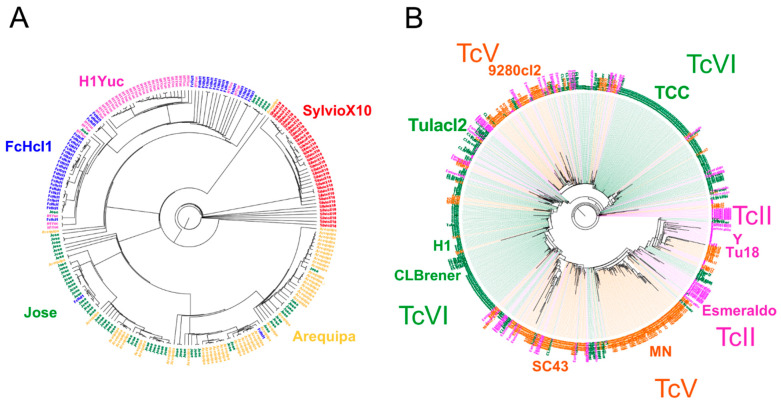
Analysis of multiple copies of mini-exon sequences. Phylogenetic analysis of Mini-exon sequences was performed for TcI (**A**) and TcII, TcV and TcVI (**B**). The names of the strains included in each tree are indicated (Arequipa, Jose, FcHcl1, H1Yuc, and SylvioX10 for TcI, and CLBrener, Tulacl2, 9280cl2, TCC, Y, Tu18, Esmeraldo, MN and SC43 for TcII, TcV and TcVI). Most copies of mini-exon sequences cluster by strain and by DTU in a Russian doll pattern, although some strains present multiple clusters and a small number of TcII sequences clustered with either TcV or TcVI sequences, in agreement with the hybrid nature of the latter DTUs.

**Figure 4 pathogens-10-00212-f004:**
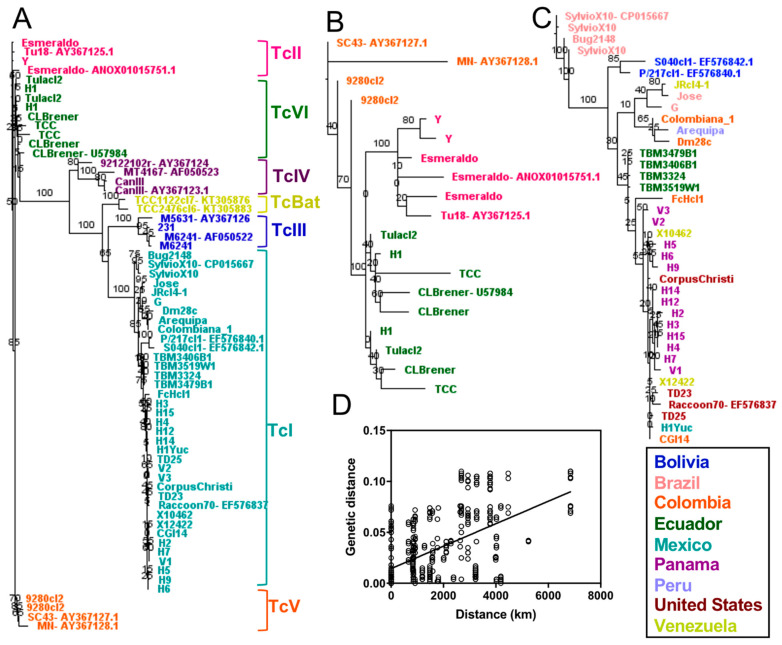
Phylogenetic analysis of a single dominant mini-exon sequence per strain. Phylogenetic analyses included all DTUs (**A**), TcII, TcV and TcVI (**B**) and TcI only (**C**). All DTUs are clearly resolved with this marker and intra-lineage diversity was particularly apparent for DTU TcI (**C**). A clear isolation by distance was observed within TcI (**D**), Mantel test (*p* = 0.0002, R = 0.544).

**Figure 5 pathogens-10-00212-f005:**
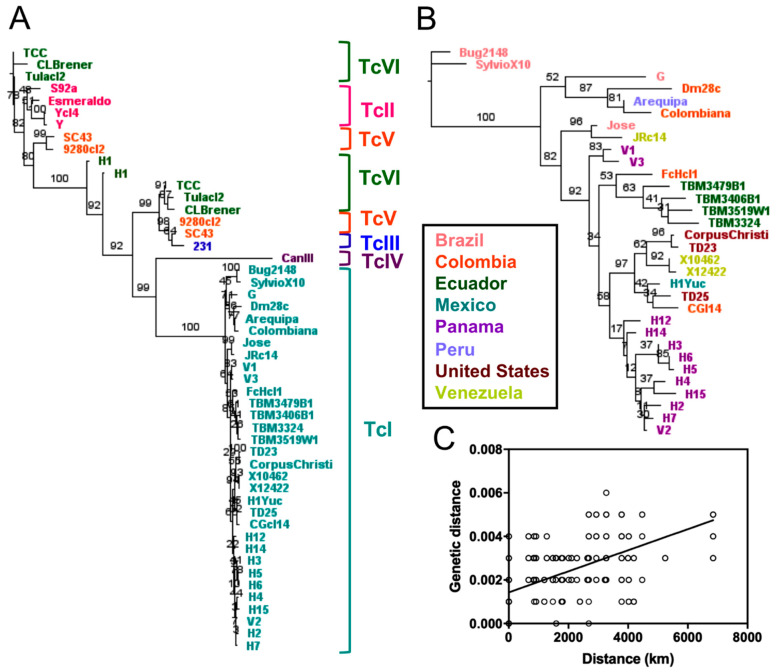
Phylogenetic trees from concatenated sequences of most available markers. *n* = 30 markers corresponding to ~26 kbp. Clear clustering by DTU was observed (**A**). A separate analysis of TcI sequences (**B**) showed multiple well-supported clusters within DTU, with diversity again mostly corresponding to geographic origin (**C**) by Mantel test, (*p* = 0.0001, R = 0.35).

**Figure 6 pathogens-10-00212-f006:**
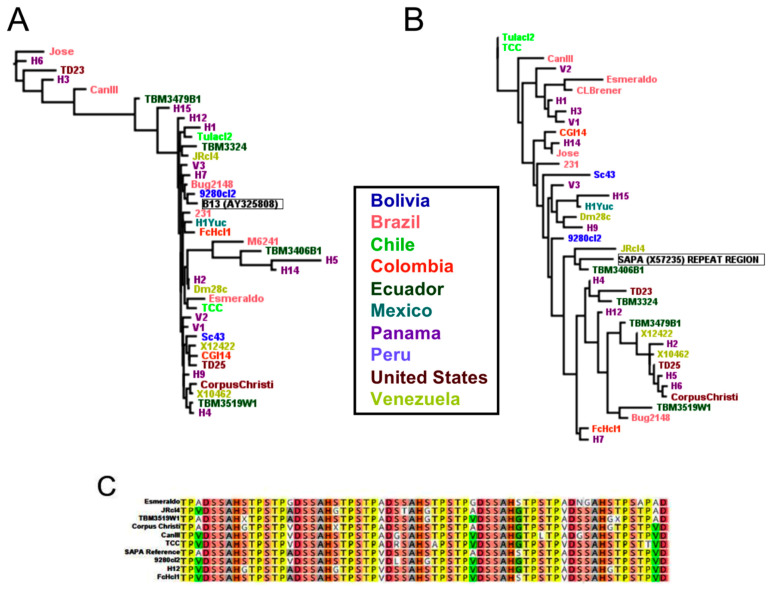
Comparison of antigen sequence diversity across strains for currently utilized diagnostic antigens. Antigen B13 (**A**) and the SAPA repeat region (**B**,**C**), demonstrated limited sequence conservation across strains, although diversity does not seem to be clearly structured according to geography.

**Table 1 pathogens-10-00212-t001:** *Trypanosoma cruzi* strains and genomes.

Strain	DTU	Accession No.	Host of Origin	Country
JRcl4	TcI	SRX290934, SRX180959, SRX180957, SRX031468, SRX031469	Human	Venezuela
Jose	TcI	SRR2057803	Human	Brazil
Colombiana	TcI	SRX871218	Human	Colombia
Arequipa	TcI	SRR1793866	Triatomine	Peru
Dm28c *	TcI	GCA_003177105.1	Opossum	Colombia
TBM3519W1	TcI	SRR3676270	Triatomine	Ecuador
TBM3324	TcI	SRR3676267	Triatomine	Ecuador
TBM3406B1	TcI	SRR3676268	Triatomine	Ecuador
TBM3479B1	TcI	SRR3676269	Triatomine	Ecuador
FcHcl1	TcI	SRR3676318	Human	Colombia
V1	TcI	SRR3676313	Triatomine	Panama
V2	TcI	SRR3676314	Triatomine	Panama
V3	TcI	SRR3676315	Triatomine	Panama
H1Yuc	TcI	SRX1851500	Human	Mexico
H1	TcVI **	SRR3676277	Human	Panama
H2	TcI	SRR3676278	Human	Panama
H3	TcI	SRR3676279	Human	Panama
H4	TcI	SRR3676280	Human	Panama
H5	TcI	SRR3676281	Human	Panama
H6	TcI	SRR3676282	Human	Panama
H7	TcI	SRR3676283	Human	Panama
H9	TcI	SRR3676285	Human	Panama
H12	TcI	SRR3676309	Human	Panama
H14	TcI	SRR3676310	Human	Panama
H15	TcI	SRR3676312	Human	Panama
TD23	TcI	SRR3676272	Triatomine	USA
TD25	TcI	SRR3676273	Triatomine	USA
Corpus Christi	TcI	SRX1054555	Human	USA
Bug2148 *	TcI **	NMZN00000000	Human	Brazil
X12422	TcI	SRR3676275	Human	Venezuela
X10462	TcI	SRR3676274	Human	Venezuela
CGl14	TcI	SRX1851527	Human	Colombia
SylvioX10	TcI	ADWP00000000	Human	Brazil
G *	TcI	MKKV00000000	Opossum	Brazil
Esmeraldo	TcII	SRX022423, SRX022425, SRX022426, SRX022427, SRX022428, SRX022429, SRX271443, SRX271444, SRX2015243, SRX2015244	Human	Brazil
S11	TcII	SRX3453751	Human	Brazil
S15	TcII	SRX3453752	Human	Brazil
S154a	TcII	SRX3453753	Human	Brazil
S23b	TcII	SRX3453755	Human	Brazil
S1162a	TcII	SRX3453754	Human	Brazil
S92a	TcII	SRX3453757	Human	Brazil
S44a	TcII	SRX3453756	Human	Brazil
Y	TcII	SRR1796718, SRR1797819	Human	Brazil
Ycl4 *	TcII	GCA_003594405.1	Human	Brazil
231 *	TcIII	OGCJ00000000	Human	Brazil
CanIII	TcIV	SRR1996498, SRR1996501	Human	Brazil
9280 cl2	TcV	SRR1996492, SRR1996493, SRR1996496, SRR1996497, SRR1996502	Human	Bolivia
SC43 *	TcV	GCA_015455285.1	Triatomine	Bolivia
CLBrener *	TcVI	GCA_000209065.1	Triatomine (culture derivate)	Brazil
Tula cl2	TcVI	SRX268890, SRX268891, SRX268892, SRX268893, SRX268894, SRX268895, SRX268896	Human	Chile
TCC *	TcVI	GCA_003177095.1	Human (culture derivate)	Chile

* Genome assemblies were used for these strains. ** Strains Bug2148 and H1 had been previously identified as belonging to TcV and TcI, respectively, but as shown in this study belong to TcI and TcVI, respectively.

**Table 2 pathogens-10-00212-t002:** Current commercial serological diagnostic antigens evaluated for sequence diversity across strains.

Antigen	Accession No.	Current Diagnostic
Ag1 [63]	M21330	Chagatest recombinant (Wiener), T-detect (InBios)
Ag36 [63]	M21331	Chagatest recombinant (Wiener), T-detect (InBios)
Ag30 [63,64]	n/a	Chagatest recombinant (Wiener), T-detect (InBios)
KMP-11 [63,65]	n/a	T-detect (InBios)
B13 [66]	AY325808	Stat-Pak (Chembio)
SAPA [63,67]	X57235	Chagatest recombinant (Wiener), T-detect (InBios)
TcH49 [66]	L09564	Stat-Pak (Chembio)

## Data Availability

All genomic sequence data for this study are available at the TriTryp (https://tritrypdb.org/tritrypdb/) and SRA (https://www.ncbi.nlm.nih.gov/sra) and GenBank (https://www.ncbi.nlm.nih.gov/genbank/) databases.

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
