# Peer review of "Assessing Trypanosoma cruzi Parasite Diversity through Comparative Genomics: Implications for Disease Epidemiology and Diagnostics"

_pathogens, 2021, doi:10.3390/pathogens10020212_

Round 1

Reviewer 1 Report

This is an excellent paper, providing valuable new insights about Trypanosoma cruzi genomic and antigenic diversity. I fully recommend it for publications.

I only have a few, although important, reservations :

Page 2 lines 74-75 : I wouldn’t be so affirmative than the authors about diploidy in T. cruzi. Some data do support widespread aneuploidy in this species, as it is the case in Leishmania :

Minning, T.A., Weatherly, D.B., Flibotte, S., Tarleton, R.L. 2011. Widespread, focal copy number variations (CNV) and whole chromosome aneuploidies in Trypanosoma cruzi strains revealed by array comparative genomic hybridization. BMC Genomics 12: 139.

Souza, R.T., Lima, F.M., Moraes Barros, R., Cortez, D.R., Santos, M.F., Cordero, E.M., Conceiçao Ruiz, J., Goldenberg, S., Teixeira, M.M.G. & Franco da Silveira, J. 2011. Genome Size, Karyotype Polymorphism and Chromosomal Evolution in Trypanosoma cruzi. Plos One 6: 1-14.

Page 2, lines 87-89 : « However, evidence of extensive sexual reproduction and recombination has recently been observed within TcI, which together with high levels of inbreeding maintained the apparent clonal structure [50,51] but may allow to generate strain diversity. ».

Page 10, lines 273-275 : « Indeed, while T. cruzi population structure has been assumed to be mostly clonal [49,62,63], recent studies pointed 274 out to frequent sexual reproduction and recombination, associated with a high inbreeding 275 [50,51] ».

I would advise editing these sentences in a more specific way.

The work by Berry et al. (2019) evidenced only that « the expectations from a purely asexually-reproducing population » were rejected.

Subjective terms such as « frequent », « extensive » sexual reproduction are poorly informative and should be replaced with clear-cut criteria. The remarkable work by Schwabl et al. (2019) did not evidence « extensive » reproduction and recombination, but rather, events of MEIOSIS that remain scarce (3 meioses/1000 mitoses), and are unable either to generate panmixia or to break the prevalent pattern of genetic clustering and of bifurcating trees, in themselves evidence for limited recombination. As a matter of fact, if « extensive » recombination events were operating, it should erase the phylogenetic signal in these populations, which is not the case. This is fully corroborated by the relevant results of the authors themselves : « Within DTUs, a strong genetic structuring of T. cruzi was also detected within TcI, in 278 agreement with a previous study » (page 10 line 278) : « strong genetic structuring » is incompatible with a pattern of « extensive recombination ». As a matter of fact, the frequence of recombination is inversely proportional to the possibility of designing bifurcating trees (Hanage,W.P., 2016. Not so simple after all: bacteria, their population genetics, and recombination.

Cold Spring Harb. Perspect. Biol. https://doi.org/10.1101/cshperspect.a018069; Maiden, M.C.J., 2006. Multilocus sequence typing of bacteria. Ann. Rev. Microbiol. 60, 561–588).

To make a long story short, the two proposals « predominant clonal evolution structure » and « extensive sexual reproduction and recombination » are exclusive of each other, unless  the authors consider that 3 meioses/1000 mitoses could be considered as « extensive genetic recombination », which would be misleading.

Now I totally agree on the fact that such occasional bouts of genetic exchange will generate genetic diversity and will have a high evolutionary relevance in the long term.

Page 6, line 115. The authors rank Bug 2148 in TCI. I fully trust their results. However, they should raise the possibility of laboratory mix-ups in the past. This strain has been identified by several authors with various techniques as TCV. It is imperative to have this strain recloned with verification of the actual cloning under the microscope.

Author Response

Reviewer #1:

Comments and Suggestions for Authors

This is an excellent paper, providing valuable new insights about Trypanosoma cruzi genomic and antigenic diversity. I fully recommend it for publications.

ANSWER: We thank the reviewer for his/her appreciation of our study.

I only have a few, although important, reservations:

Page 2 lines 74-75: I wouldn’t be so affirmative than the authors about diploidy in T. cruzi. Some data do support widespread aneuploidy in this species, as it is the case in Leishmania: 

Minning, T.A., Weatherly, D.B., Flibotte, S., Tarleton, R.L. 2011. Widespread, focal copy number variations (CNV) and whole chromosome aneuploidies in Trypanosoma cruzi strains revealed by array comparative genomic hybridization. BMC Genomics 12: 139.

Souza, R.T., Lima, F.M., Moraes Barros, R., Cortez, D.R., Santos, M.F., Cordero, E.M., Conceiçao Ruiz, J., Goldenberg, S., Teixeira, M.M.G. & Franco da Silveira, J. 2011. Genome Size, Karyotype Polymorphism and Chromosomal Evolution in Trypanosoma cruzi. Plos One 6: 1-14.

ANSWER: We agree with the reviewer that although T. cruzi is mostly described as diploid, aneuploidy seems to be widespread. We have edited this sentence to clarify this point as suggested (page 2).

Page 2, lines 87-89: « However, evidence of extensive sexual reproduction and recombination has recently been observed within TcI, which together with high levels of inbreeding maintained the apparent clonal structure [50,51] but may allow to generate strain diversity. ».

Page 10, lines 273-275: « Indeed, while T. cruzi population structure has been assumed to be mostly clonal [49,62,63], recent studies pointed 274 out to frequent sexual reproduction and recombination, associated with a high inbreeding 275 [50,51] ».

I would advise editing these sentences in a more specific way. The work by Berry et al. (2019) evidenced only that « the expectations from a purely asexually-reproducing population » were rejected. Subjective terms such as « frequent », « extensive » sexual reproduction are poorly informative and should be replaced with clear-cut criteria. The remarkable work by Schwabl et al. (2019) did not evidence « extensive » reproduction and recombination, but rather, events of MEIOSIS that remain scarce (3 meioses/1000 mitoses), and are unable either to generate panmixia or to break the prevalent pattern of genetic clustering and of bifurcating trees, in themselves evidence for limited recombination. As a matter of fact, if « extensive » recombination events were operating, it should erase the phylogenetic signal in these populations, which is not the case. This is fully corroborated by the relevant results of the authors themselves : « Within DTUs, a strong genetic structuring of T. cruzi was also detected within TcI, in 278 agreement with a previous study » (page 10 line 278) : « strong genetic structuring » is incompatible with a pattern of « extensive recombination ». As a matter of fact, the frequence of recombination is inversely proportional to the possibility of designing bifurcating trees (Hanage,W.P., 2016). Not so simple after all: bacteria, their population genetics, and recombination.

Cold Spring Harb. Perspect. Biol. https://doi.org/10.1101/cshperspect.a018069; Maiden, M.C.J., 2006. Multilocus sequence typing of bacteria. Ann. Rev. Microbiol. 60,561–588).

To make a long story short, the two proposals « predominant clonal evolution structure » and « extensive sexual reproduction and recombination » are exclusive of each other, unless the authors consider that 3 meioses/1000 mitoses could be considered as « extensive genetic recombination », which would be misleading. Now I totally agree on the fact that such occasional bouts of genetic exchange will generate genetic diversity and will have a high evolutionary relevance in the long term.

ANSWER: We agree with the reviewer’s point as this is indeed a complex issue, and we have reworded these sentences to be more specific as suggested (pages 2 &10).

Page 6, line 115. The authors rank Bug 2148 in TCI. I fully trust their results. However, they should raise the possibility of laboratory mix-ups in the past. This strain has been identified by several authors with various techniques as TCV. It is imperative to have this strain recloned with verification of the actual cloning under the microscope.

ANSWER: We thank the reviewer for this comment. We do agree that a laboratory mix-up is a possible explanation for the difference in results, as several previous studies clearly genotyped this strain as TcV. This is suggested in the discussion (top of page 11).

Reviewer 2 Report

In the manuscript with the title “Assessing Trypanosoma cruzi parasite diversity through comparative genomics: implications for disease epidemiology and diagnostics” by Majeau A. et al., the authors perform the phylogenetic analysis of 52 sequenced T. cruzi strains based on their genomes available at the databases. They claim that phylogenetic analysis and analysis based on 32 markers clustered the sequences by DTU. Also, they conclude that the diversity of antigens used in diagnostic can explain the limited diagnostic performance. Thus, the article is potentially interesting to the field, however the way it was performed needs improvement.

Despite it is a good idea to give some light on the complex parasite genome, the authors did not include all the genomes available in their analysis, not even the last improved versions of some of them. Thus, they included in the phylogenetic analysis old versions of the genomes that, due to the different methodologies used, are of variable quality, being more fragmented than the more recent ones, which can bias the conclusions.

Besides, not all 52 strains are used in all the phylogenetic analyses. In figure 1 almost all the current genomes found in TritrypDB are missing and should be included. The reference strain CL Brener is missing in Fig 1.

Moreover, in figure 3, where they analyze multiple copies of mini-exon obtaining a Russian doll pattern, it is not clear which genome strains were taken into account. Amplifying the figure, I could not see whether Bug2148 was included. Also, it looks like not all the strains were analyzed together, since TcI strains are represented apart from the rest.

Also, I wonder whether the number of genomes of each DTU can influence the phylogenetic analysis, there are almost 10 times more TcI than the rest of the other DTUs. They discuss this, but it likely biases the results.

Thus, I suggest the authors include the latest versions of the genomes in all the analysis, whenever data is available.

Concerning the diversity of the antigens, in figure 6 they align the predicted protein sequence of SAPA from different strains, including, as I understand, only the 5 repeats used in diagnosis, while SAPA repeats are not identical in the same strain, and in more than 10 repeats which can align in different ways. Thus the question, in this case, is whether the other repeats are similar to what is used in diagnosis. I also suggest using better tools for the alignments instead of BLAST.

Author Response

Reviewer #2:

In the manuscript with the title “Assessing Trypanosoma cruzi parasite diversity through comparative genomics: implications for disease epidemiology and diagnostics” by Majeau A. et al., the authors perform the phylogenetic analysis of 52 sequenced T. cruzi strains based on their genomes available at the databases. They claim that phylogenetic analysis and analysis based on 32 markers clustered the sequences by DTU. Also, they conclude that the diversity of antigens used in diagnostic can explain the limited diagnostic performance. Thus, the article is potentially interesting to the field, however the way it was performed needs improvement. Despite it is a good idea to give some light on the complex parasite genome, the authors did not include all the genomes available in their analysis, not even the last improved versions of some of them. Thus, they included in the phylogenetic analysis old versions of the genomes that, due to the different methodologies used, are of variable quality, being more fragmented than the more recent ones, which can bias the conclusions.

ANSWER: We thank the reviewer for this comment. It is correct that a couple of parasite genomes have become available after we started our analyses, and we can expect that new ones will regularly become available in the future. Nonetheless, we believe that the set of genomes included in our analysis is the most exhaustive to date as it covers most of T. cruzi DTU diversity. Adding one or two genomes to our analysis is unlikely to affect our general conclusions, but as we clearly discuss, it will be key to expand these analyses once multiple additional genomes become available from some underrepresented geographic areas and DTUs.

We are also aware there are differences in genome quality and coverage due to different methodologies, which is why we reassembled many of these genomes in order to improve the assemblies of some of the older genomes and have a more homogenous dataset that minimized potential bias from sequencing methodologies. This is now pointed out in the Methods section (page 13, top). In any case, such bias would be absent from the analyses of the selected markers as we excluded markers from genomes with poor sequence coverage.

Besides, not all 52 strains are used in all the phylogenetic analyses. In figure 1 almost all the current genomes found in TritrypDB are missing and should be included. The reference strain CL Brener is missing in Fig 1.

ANSWER: It is correct that not all genomes were included in all analyses as we selected representative subsets covering in a more balanced manner a wide geographic distribution and DTU diversity, with the best genome coverage. For example, the CL Brener reference genome was excluded from Fig. 1A due to its limited coverage/assembly compared to the other TcVI genomes that were included. As mentioned above, this selection minimized bias due to differences in genome methodologies/assemblies.

Moreover, in figure 3, where they analyze multiple copies of mini-exon obtaining a Russian doll pattern, it is not clear which genome strains were taken into account. Amplifying the figure, I could not see whether Bug2148 was included. Also, it looks like not all the strains were analyzed together, since TcI strains are represented apart from the rest.

ANSWER: Only parasite strains with sufficient genome coverage allowing to identify large numbers of mini exon paralogous sequences (see supplementary table 2) were included in Fig. 3. The strain names are indicated in the figure, but they have now been added in the figure legend as well for greater clarity. TcI and TcII-TcV-TcVI mini exon sequences were indeed analyzed separately because of large sequence differences among these DTUs for this marker.

Also, I wonder whether the number of genomes of each DTU can influence the phylogenetic analysis, there are almost 10 times more TcI than the rest of the other DTUs. They discuss this, but it likely biases the results.

Thus, I suggest the authors include the latest versions of the genomes in all the analysis, whenever data is available.

ANSWER: We appreciate the reviewer’s comment, and this is why we performed separate analyses for TcI strains, which are indeed over-represented. These separate analyses show that there is important diversity within TcI DTU, but a lot of this diversity is lost when multiple DTUs are analyzed together (for example compare Fig. 1A and 1B, Fig. 4A and 4C, or Fig. 5A and 5B). Some similar diversity within DTUs is also emerging when analyzing TcII, TcV and TcVI DTUs separately, in spite of a small number of strains. Hence the interest of presenting analyses with different sets of DTUs.

Concerning the diversity of the antigens, in figure 6 they align the predicted protein sequence of SAPA from different strains, including, as I understand, only the 5 repeats used in diagnosis, while SAPA repeats are not identical in the same strain, and in more than 10 repeats which can align in different ways. Thus the question, in this case, is whether the other repeats are similar to what is used in diagnosis. I also suggest using better tools for the alignments instead of BLAST.

ANSWER: We thank the reviewer for this comment. Both the full length SAPA sequence and the repeat regions were analyzed. The full length SAPA sequence analysis is available in supplementary Figure 2 and the analysis of the repeat region in Figure 6B and 6C. While BLAST was used to identify SAPA sequences (as well as other antigens) in each genome, alignments were constructed using Muscle and phylogenies constructed with PHYML, as indicated in the methods section (page 15).

Reviewer 3 Report

The authors give an interesting overview of the relationship of different trypanosome strains at different genetic levels and use adequate methods here. The authors find great similarities between the different methods of analysis. The relationships also reflect the regional sources with some exceptions. In addition the authors  were able to correct the type affiliation for different strains (e.g. H1).
The work is based on a large number of strains from different regions of the Americas, covering the major groups.
In addition, the authors address the problem of diagnostics and provide evidence as to why diagnostic tests work more or less well depending on the region. These important clues can be used to improve the tests depending on the geographic region.
In the discussion the possible starting points for future further research results are laid to investigate the relationship in more detail. For example, the number of type III and type IV strains is relatively small, so further analysis is of interest here.

Overall, there is little cause for criticism. I would like to mention only a few points:
Fig. 1 (B), what could be the cause of the segregation of strains from one country, e.g. Venezuela with strains (Typ I) in different clusters (JRcl4),  (X12422, X10462): different regions, different sources? This could shortly discussed.

Fig. 1C, 4D, 5C: what is the origin for the distance (km), the localization of 0 km?

Line 136: The authors mention a genetic structuration within TcI. It should be discussed in more details if such a structuration is also present in the other DTUs.

Fig. 2 for easier finding the cases of introgression should be marked in (A), perhaps by underlining oder with asterisks.

line 163 pp.: the difference between flagellum-adhesion glycoprotein or beta-adaptin to the other proteins for intra-lineage variation can not be clearly seen in the supplemental figure. Here the authors should describe the peculiarity of these proteins in more detail.

line 189, correct the reference to Figure 4D (wrong: 3D).

linr 281, correct the word to sublineages (wrong: lineajes)

Host influence should be discussed in the context of regional distributions of strains. Thus, people can travel and animals can migrate. That means, how clear is it that a specific strain is endemic and has not been introduced by people or that people have become infected somewhere else than where they live and have been sampled.

In addition it would be of interest to me  how the outer group, Trypanosoma rangeli, a relative to T. cruzi, but apathogenic, fits into the phylogenetic tree of whole genome analyses. This could perhaps give some hints how and when T. cruzi and T. rangeli diverged and if the ancestors of both were parasitic or not.

Author Response

Reviewer #3:

Comments and Suggestions for Authors

The authors give an interesting overview of the relationship of different trypanosome strains at different genetic levels and use adequate methods here. The authors find great similarities between the different methods of analysis. The relationships also reflect the regional sources with some exceptions. In addition the authors were able to correct the type affiliation for different strains (e.g. H1).
The work is based on a large number of strains from different regions of the Americas, covering the major groups.
In addition, the authors address the problem of diagnostics and provide evidence as to why diagnostic tests work more or less well depending on the region. These important clues can be used to improve the tests depending on the geographic region.
In the discussion the possible starting points for future further research results are laid to investigate the relationship in more detail. For example, the number of type III and type IV strains is relatively small, so further analysis is of interest here.

Overall, there is little cause for criticism.

ANSWER: We thank the reviewer for his/her appreciation of our study.

I would like to mention only a few points:
Fig. 1 (B), what could be the cause of the segregation of strains from one country, e.g. Venezuela with strains (Typ I) in different clusters (JRcl4),  (X12422, X10462): different regions, different sources? This could shortly discussed.

ANSWER: As suggested, a sentence was added to the discussion shortly discussing this. Indeed, different clusters from the same country may result from distinct transmission cycles with different hosts and environments at a more local level than country (page 10).

Fig. 1C, 4D, 5C: what is the origin for the distance (km), the localization of 0 km?

ANSWER: Pair-wise geographic distance among parasite strains were calculated based on country of origin, so strains from the same country have a 0 km geographic distance. This is now specified in the methods section (page 14).

Line 136: The authors mention a genetic structuration within TcI. It should be discussed in more details if such a structuration is also present in the other DTUs.

ANSWER: We mention in the discussion that our analyses suggest intra-lineage diversity within TcII and TcVI, though we recognize that more genomes must be analyzed for all non-TcI DTUs to firmly draw this conclusion (Page 10).

Fig. 2 for easier finding the cases of introgression should be marked in (A), perhaps by underlining oder with asterisks.

ANSWER: As suggested, an arrow has been added to figure 2A to mark the cases of introgression.

line 163 pp.: the difference between flagellum-adhesion glycoprotein or beta-adaptin to the other proteins for intra-lineage variation can not be clearly seen in the supplemental figure. Here the authors should describe the peculiarity of these proteins in more detail.

ANSWER: We did not intend to provide extensive comparisons of the different markers we used, and just provided a general qualitative assessment of their ability to discriminate among DTUs and to detect within DTU diversity.

line 189, correct the reference to Figure 4D (wrong: 3D).

ANSWER: Thank you for catching this error. The text has been corrected.

line 281, correct the word to sublineages (wrong: lineajes)

ANSWER: Corrected.

Host influence should be discussed in the context of regional distributions of strains. Thus, people can travel and animals can migrate. That means, how clear is it that a specific strain is endemic and has not been introduced by people or that people have become infected somewhere else than where they live and have been sampled.

ANSWER: We thank the reviewer for this comment. We agree that introductions of strains due to host movement to an area may, in part, explain some of the differences in clustering for different strains from the same country. This has been added to the discussion (page 10).

In addition, it would be of interest to me how the outer group, Trypanosoma rangeli, a relative to T. cruzi, but apathogenic, fits into the phylogenetic tree of whole genome analyses. This could perhaps give some hints how and when T. cruzi and T. rangeli diverged and if the ancestors of both were parasitic or not.

ANSWER: We appreciate the reviewer’s comment and agree that understanding the divergence between T. rangeli and T. cruzi would be interesting, but we believe that this is beyond the scope of this study. In addition, previous studies with different markers suggest that including this outgroup can result in the collapse of T. cruzi diversity, which is the scope of this paper (See for example Bradwell et al., BMC Genomics (2018) 19:770).

Reviewer 4 Report

In this manuscript, the authors describe genomic analysis of 52 previously published T. cruzi genomes from a variety of DTUs and geographic locations. At the whole genome level there was clear clustering according to DTU and sub clustering within TcI with some link to geographical location. Further analysis was performed using single and multiple markers, whilst all markers showed similar diversity between the DTUs, only a small number showed intra-linage variation. Furthermore, the authors investigate 7 antigens currently used in current diagnostic tests, three were found to be poorly conserved across the T. cruzi genomes and the reference antigens currently used differed from sequences within other stains.

Broad comments:

Overall, this is a good quality paper. The authors provide sufficient information within the introduction describing the reasoning behind and the scope of the study.

The results are presented well and are easy to read and follow and they provide evidence for all conclusions drawn. I found it a compelling and interesting read.

Specific comments:

Line 56 could reference Miles at al 1981, Lancet - first mention of link between differing geographical distribution of T. cruzi DTUs and different clinical manifestations.

In the introduction they mention 7 possible DTUs including TcI-TcVI and TcBat, however TcBat does not appear to be included in the geonomic analysis only appearing in one of the phylogenetic trees - it may be useful, to the reader, to describe why this. 

Author Response

Reviewer #4:

Comments and Suggestions for Authors

In this manuscript, the authors describe genomic analysis of 52 previously published T. cruzi genomes from a variety of DTUs and geographic locations. At the whole genome level there was clear clustering according to DTU and sub clustering within TcI with some link to geographical location. Further analysis was performed using single and multiple markers, whilst all markers showed similar diversity between the DTUs, only a small number showed intra-linage variation. Furthermore, the authors investigate 7 antigens currently used in current diagnostic tests, three were found to be poorly conserved across the T. cruzi genomes and the reference antigens currently used differed from sequences within other stains.

Broad comments:

Overall, this is a good quality paper. The authors provide sufficient information within the introduction describing the reasoning behind and the scope of the study.

The results are presented well and are easy to read and follow and they provide evidence for all conclusions drawn. I found it a compelling and interesting read.

ANSWER: We thank the reviewer for his/her appreciation of our study.

Specific comments:

Line 56 could reference Miles at al 1981, Lancet - first mention of link between differing geographical distribution of T. cruzi DTUs and different clinical manifestations.

ANSWER: We thank the reviewer for their suggestion and have added the Miles et al. 1981 reference to this section.

In the introduction they mention 7 possible DTUs including TcI-TcVI and TcBat, however TcBat does not appear to be included in the geonomic analysis only appearing in one of the phylogenetic trees - it may be useful, to the reader, to describe why this. 

ANSWER: We thank the reviewer for this observation and have added text to the Introduction (page 1, bottom) and the Methods (page 11) sections to clarify that no whole genome of TcBat is available, as more extensive molecular studies are yet needed for this more newly described DTU.

Round 2

Reviewer 2 Report

The authors provided adequate responses to some of the queries, but the manuscript still lacks relevant information and discussion. Because of that I recommend reconsidering after revision.

Regarding figure 1, please give details on which genomes were reassembled by the authors, and include this information in Table 2. Will you make these new assemblies be available to the public?

I would like to point out that comparative genomics is more robust than selected markers analysis, and it is the main contribution of the manuscript. Thus, the authors should consider other explanations for the strains which cluster in a different DTU than originally based on selected markers. Despite most of the strains clustered in the original DTU, it could be that the discrepant strains, because of a higher coverage are more similar to other DTUs. Please argue in the discussion section. The phylogenetic analysis could unravel more genomic variability than previously thought and could eventually explain different virulence among strains belonging to the same DTU, for which there is no explanation to date.

Regarding figure 3, the answer of the authors raised some questions:

Why in figure 3 Bug2148 which has 173 copies according to supplementary Table 2 was not included? Please explain
Why only 5 TcI were taken into account for mini-exon analysis, while there were many with similar copy number. What is the purpose of such analysis that excludes so many strains? Please explain.

Please correct CL Brenner to CL Brener in Supplementary Table 2.

Author Response

Comments and Suggestions for Authors

The authors provided adequate responses to some of the queries, but the manuscript still lacks relevant information and discussion. Because of that I recommend reconsidering after revision.

ANSWER: We thank the reviewer for these further comments.

Regarding figure 1, please give details on which genomes were reassembled by the authors, and include this information in Table 2. Will you make these new assemblies be available to the public?

ANSWER: Please see page 13 in the methods section and in Table 1 for the list of genomes for which current assemblies were used and those that were newly reassembled. These assemblies are available upon request.

I would like to point out that comparative genomics is more robust than selected markers analysis, and it is the main contribution of the manuscript. Thus, the authors should consider other explanations for the strains which cluster in a different DTU than originally based on selected markers. Despite most of the strains clustered in the original DTU, it could be that the discrepant strains, because of a higher coverage are more similar to other DTUs. Please argue in the discussion section. The phylogenetic analysis could unravel more genomic variability than previously thought and could eventually explain different virulence among strains belonging to the same DTU, for which there is no explanation to date.

ANSWER: We thank the reviewer for this comment. We certainly agree that the whole genome analysis is more robust than any single marker analysis. Nonetheless, our analyses comparing phylogenies based on different markers to the whole genome phylogenies demonstrate that these markers do, in fact, fairly well recapitulate the whole genome diversity. In particular for the discrepant strains, DTU clustering was identical for all markers and whole genomes, providing evidence against increased sequence resolution being a likely explanation for strain reclassification. We thus believe that a laboratory mix-up of strains/DNA is the most likely explanation for these discrepancies. This is explained in the discussion (page 11).

Regarding figure 3, the answer of the authors raised some questions:
Why in figure 3 Bug2148 which has 173 copies according to supplementary Table 2 was not included? Please explain.

Why only 5 TcI were taken into account for mini-exon analysis, while there were many with similar copy number. What is the purpose of such analysis that excludes so many strains? Please explain.

ANSWER: We thank the reviewer for this precise comment. To further clarify this point, we would like to stress that while multiple copies of the mini-exon sequences could be identified in some genomes, many were truncated sequences (likely due to limited coverage of the genomes), based on our criteria of identifying sequences as short as 250 bp. These short sequences, corresponding to different regions of the complete mini-exon (600-700 bp), prevent a rigorous sequence alignment and phylogenetic analysis. We thus had to compromise based on both copy number and sequence length, which is why only these 5 TcI strains were included in Figure 3A. This is now better explained in the methods section (Page 14).

Please correct CL Brenner to CL Brener in Supplementary Table 2.

ANSWER: Thank you for catching this error. The text has been corrected.

Round 3

Reviewer 2 Report

I am satisfied with the answer to my comments.